# Observation of vortex-string chiral modes in metamaterials

Jingwen Ma [1], Ding Jia[2,3], Li Zhang[2,4,5,6], Yi-jun Guan[3], Yong Ge[3], Hong-xiang Sun [3,7] ✉, Shou-qi Yuan [3], Hongsheng Chen [2,4,5,6], Yihao Yang [2,4,5,6] ✉ & Xiang Zhang [1] ✉

As hypothetical topological defects in the geometry of spacetime, vortex strings could have played many roles in cosmology, and their distinct features can provide observable clues about the early universe's evolution. A key feature of vortex strings is that they can interact with Weyl fermionic modes and support massless chiral-anomaly states along strings. To date, despite many attempts to detect vortex strings in astrophysics or to emulate them in artificially created systems, observation of these vortex-string chiral modes remains experimentally elusive. Here we report experimental observations of vortex-string chiral modes using a metamaterial system. This is implemented by inhomogeneous perturbation of Yang-monopole phononic metamaterials. The measured linear dispersion and modal profiles confirm the existence of topological modes bound to and propagating along the string with the chiral anomaly. Our work provides a platform for studying diverse cosmic topological defects in astrophysics and offers applications as topological fibres in communication techniques.

Topology, originally a mathematical subject describing the conserved global quantities of geometric objects, now plays an indispensable role in various aspects of modern physics. In high-energy physics, topological defects, including Weyl, Dirac, and Yang monopoles[1–3], are powerful handles to describe various elementary particles. For instance, Dirac fermions describe the behaviour of electrons and quarks in the Standard Model, and Weyl fermions were once thought to describe neutrinos for a long time until the discovery of neutrino oscillation. Recently, artificially engineered systems such as cold atoms[4], condensed matters[5], and photonic or phononic metamaterials[6,7] have emerged as versatile platforms for the observation of the low-energy counterparts of these topological phenomena[8–13]. Realisations of Weyl and Dirac monopoles in crystalline lattices have unveiled intriguing

phenomena that are otherwise experimentally inaccessible in high-energy physics, such as surface Fermi arcs[8,14], chiral anomaly[15], and Dirac hierarchy[16]. The Dirac monopoles can also be generalised to the so-called Yang monopoles (YM) in five dimensions, which have been investigated in cold-atom and photonic systems, leading to experimental observations of non-Abelian second Chern numbers[17] and linked Weyl surfaces/arcs[18].

Among many fascinating topological phenomena, vortex strings (or cosmic strings)[19,20] stand out as a typical object that has attracted broad interest in many areas of physics. Vortex strings are postulated to arise from the spontaneous symmetry breaking of the Higgs field during the early stages of the universe. They have been initially proposed as a potential candidate that could play a role in the formation of

[1]Faculties of Science and Engineering, The University of Hong Kong, Hong Kong, China. [2]Interdisciplinary Center for Quantum Information, State Key Laboratory of Extreme Photonics and Instrumentation, ZJU-Hangzhou Global Scientific and Technological Innovation Center, Zhejiang University, Hangzhou 310027, China. [3]Research Center of Fluid Machinery Engineering and Technology, School of Physics and Electronic Engineering, Jiangsu University, Zhenjiang 212013, China. [4]International Joint Innovation Center, The Electromagnetics Academy at Zhejiang University, Zhejiang University, Haining 314400, China. [5]Key Laboratory of Advanced Micro/Nano Electronic Devices & Smart Systems of Zhejiang, Jinhua Institute of Zhejiang University, Zhejiang University, Jinhua 321099, China. [6]Shaoxing Institute of Zhejiang University, Zhejiang University, Shaoxing 312000, China. [7]State Key Laboratory of Acoustics, Institute of Acoustics, Chinese Academy of Sciences, Beijing 100190, China. ✉e-mail: jsdxshx@ujs.edu.cn; yangyihao@zju.edu.cn; president@hku.hk

galaxies in our cosmos[21]. Continuous efforts have been made to observe vortex strings and set new constraints on their energy scales[22–24], providing valuable insights into the potential existence of these cosmic structures. While vortex strings have not yet been observed in astrophysics, similar concepts can be emulated in low-energy systems, including type-II superconductors[25], liquid helium[26], and liquid crystals[27]. In 1985, Edward Witten made a significant theoretical prediction about vortex strings[28]: when interacting with Weyl fermionic fields, vortex strings can support chiral-anomaly propagating modes with massless dispersions. Such chiral modes can support the directional flow of charged fermions without external electrical fields, making the vortex strings act as "superconducting strings", which might provide observable clues about the evolution of the early universe[29]. More recently, this key feature of vortex strings – the massless vortex-string chiral (VSC) mode – has been theoretically extended to condensed matter systems[30–33] and photonic metamaterials[34], leading to a promising avenue to realise helical Majorana fermions[35] and topological fibres[36] respectively. However, experimental observation of such massless VSC modes remains elusive.

Here, we experimentally observe the VSC modes in metamaterials. Specifically, we apply an inhomogeneous perturbation on the YM phononic metamaterial, so that it can support a pair of VSC modes with linear gapless dispersion and chiral anomaly propagation behaviour. Using pump-probe measurements, we observe the linear gapless dispersion, the modal profiles tightly confined to core of vortex string, and the chiral anomaly flow of acoustic waves along it. Our findings not only provide a conclusive experimental advance on the observation of VSC modes but also open the possibility of constructing high-performance and robust fibres for future information technologies.

## Results

### Theoretical description of YM and VSC modes

As shown in Fig. 1a, we begin with the discussion of a single YM[3]. It contains a pair of Weyl monopoles defined in three-dimensional (3D) momentum space $(k_1, k_2, k_3)$ with opposite first Chern numbers $C_1 = \pm 1$. The presence of a nonzero Higgs field $\Delta$, which is a complex scalar field $\Delta = k_4 + ik_5$, can mix the Weyl monopoles and delegate a nonzero mass to them[33]. In our experiment, this complex scalar field $\Delta$ is introduced by carefully varying the geometry of an ideal Weyl-monopole metamaterials with two independent synthetic parameters $k_4$ and $k_5$ (Supplementary Note Sec. 1). By considering $(k_4, k_5)$ as two synthetic momentum components, the Weyl monopole pair can be upgraded to a YM living in a five-dimensional (5D) space defined by $\vec{k} = (k_1, k_2, k_3, k_4, k_5)$, where $(k_1, k_2, k_3)$ describes the conventional momentum components along $x$, $y$, and $z$ directions. A YM is described by a generalised 4×4 Dirac Hamiltonian: $\mathbf{H} = \omega_{YM}\mathbf{I} + \sum_{n=1}^{5} v_n (k_n - k_n^{YM})\mathbf{\Gamma}_n$, where $\mathbf{I}$ is a 4×4 unitary matrix, $\mathbf{\Gamma}_n$ ($n = 1–5$) represents five 4 × 4 Gamma matrices satisfying the anticommutation relation $\{\mathbf{\Gamma}_n, \mathbf{\Gamma}_m\} = 2\delta_{mn}$, $v_n$ ($n = 1–5$) is the Dirac velocity, $\omega_{YM}$ is the energy of the YM, and $\vec{k}_{YM} = (k_1^{YM}, k_2^{YM}, k_3^{YM}, k_4^{YM}, k_5^{YM})$ describes location of YM in 5D space. Such YM features a SU(2) gauge field in 5D space[3]. Consequently, it exhibits a non-Abelian second Chern number $C_2^{YM}$ calculated by integrating the curvature of this SU(2) gauge field throughout the four-dimensional (4D) manifold enclosing the YM (Fig. 1a and Supplementary Note Section 1.3).

In cosmology, cosmic strings emerge at spacetime positions where the Higgs field frustrates along a line. Similarly, a vortex string can be implemented in metamaterials when the complex scalar field $\Delta$ exhibits nonzero vorticity and frustrates along a one-dimensional (1D) string extending in the spatial domain. As shown in Fig. 1, this can be implemented by mapping the spatial-domain cylinder coordinates $\vec{r} = (R, \phi, z)$ to the synthetic subspace $(k_4, k_5)$ so that the scalar field $\Delta(\vec{r}) = k_4 + ik_5 = \Delta_0 e^{in\phi + i\theta_0}$ changes the phase by $2n\pi$ surrounding the strings[28]. Note that $\Delta(\vec{r})$ vanishes at the origin point $R = 0$. Here, $n$

represents the vorticity of the vortex string, and $\theta_0$ is the phase of scalar field $\Delta$ at $\phi = 0$. For arbitrary values of $n$, the vortex string always supports $n$-fold degenerate topological chiral modes[36] protected by the second Chern number of vortex strings $C_2$ defined in the 4D manifold $(k_1, k_2, k_3, \phi)$. This second Chern number of vortex strings originates from the non-Abelian topological charge of YM and satisfies $C_2 = nC_2^{YM}$. Without losing any generality, we focus on the case of $n = 1$ and $C_2^{YM} = \pm 1$ in our experiments. In this case, there exist topological VSC modes with chiral wavefunction and linear dispersion (Fig. 1b, c and Supplementary Notes Section 1.4) for non-Abelian topological charge $C_2 = \pm 1$:

$$\begin{cases} \psi_- = e^{ik_3 z - \Delta_0 R}(e^{i\theta_0/2} \quad 0 \quad 0 \quad e^{-i\theta_0/2})^T, \omega(k_3) = \omega_{YM} + v_D k_3, \\ \psi_+ = e^{-ik_3 z - \Delta_0 R}(0 \quad e^{i\theta_0/2} \quad e^{-i\theta_0/2} \quad 0)^T, \omega(k_3) = \omega_{YM} - v_D k_3. \end{cases} \quad (1)$$

Here the charge $C_2 = -1$ ($C_2 = 1$) VSC mode propagates in forward (backward) directions.

### Experimental observations of VSC modes in a metamaterial

To experimentally realise the VSC modes, we design a phononic metamaterial supporting ideal YMs at K and K' points (Supplementary Fig. 3). Such YM metamaterials are realized by modifying a previously studied Weyl-monopole metamaterial[13,14] (Supplementary Figs. 2 and 3) using a Brillouin-zone-folding method (Supplementary Notes Section 1). As shown in Supplementary Fig. 3a, the metamaterial exhibits a hexagonal lattice in the $x$-$y$ plane with a lattice constant of $a_0 = 41$ mm and is periodic along the $z$-direction with a lattice constant of $h = 32$ mm. Each unit cell contains two substructures A (0 < $z$ < $h/2$) and B ($h/2$ < $z$ < $h$) with the same geometry, so that the entire geometry preserves a half-lattice translational symmetry along the $z$ direction. Each substructure contains six spiral air channels connecting the planar air layers with fan-shaped scatters. The metamaterial supports ideal Dirac monopoles (i.e. iso-frequency Weyl monopole pairs with opposite first Chern numbers) at K/K' point in Fig. 1d. These Dirac monopoles can be further promoted to YM in 5D space if we introduce two additional parameters $k_4$ and $k_5$ to break the half-lattice translational symmetry along the $z$. This is realized when the substructures A and B (marked in red and blue in Supplementary Fig. 3e) have different geometries defined by parameters $(k_4, k_5)$, where $k_4$ shrinks (expands) the size of the spiral air channels to $\alpha_0 \mp \alpha(k_4)$ and $w_0 \mp w(k_4)$ and $k_5$ rotates the fan-shape scatters counterclockwise (clockwise) by $\beta(k_5) = \mathrm{asin}(k_5)/3$. Based on tight-binding approximation[37], we theoretically confirm that such metamaterials exhibit a generalised 4×4 Dirac Hamiltonian: $\mathbf{H} = \omega_{YM}\mathbf{I} + \sum_{n=1}^{5} v_D(k_n - k_n^{YM})\mathbf{\Gamma}_n$ (Supplementary Note Sec. 1 and Figs. S1–3). The numerically calculated band diagrams projected in the $(k_1, k_2, k_3)$ momentum space with $(k_4, k_5) = (k_4^{YM}, k_5^{YM}) = (0, 0)$ (Fig. 1d) and in the $(k_4, k_5)$ synthetic space with $(k_1, k_2, k_3) = (k_1^{YM}, k_2^{YM}, k_3^{YM}) = (\pm 4\pi/3a_0, 0, 0)$ (Fig. 1e) confirm that the proposed metamaterial can support YMs at $\vec{k}_{YM} = (k_1^{YM}, k_2^{YM}, k_3^{YM}, k_4^{YM}, k_5^{YM}) = (\pm 4\pi/3a_0, 0, 0, 0, 0)$. Due to the time-reversal invariance of the system, the YMs located at the K and K' points naturally exhibit opposite second Chern numbers $C_2^{YM} = \pm 1$.

The experimentally realised phononic metamaterial device is shown in Fig. 2. The device is the opposite structure of the air region shown in Supplementary Fig. 3 and is fabricated by a 3D printing technique with photopolymer materials. The full dimension of the device is $426 \times 492 \times 576$ mm, containing 18 periods along the $z$ direction. Vortex string is implemented by mapping the spatial cylinder coordinates $\vec{r} = (R, \phi, z)$ to the synthetic space $(k_4, k_5)$, so that each unit cells exhibit different values of complex scalar field $\Delta(\vec{r}) = k_4 + ik_5 = \Delta_0 e^{in\phi + i\theta_0}$ (Fig. 2d), and the entire device is inhomogeneous in the $x$-$y$ plane. Specifically, the vorticity of the vortex string is $n = 1$, and the phase of the complex scalar field $\Delta(\vec{r})$ at $\phi = 0$ is

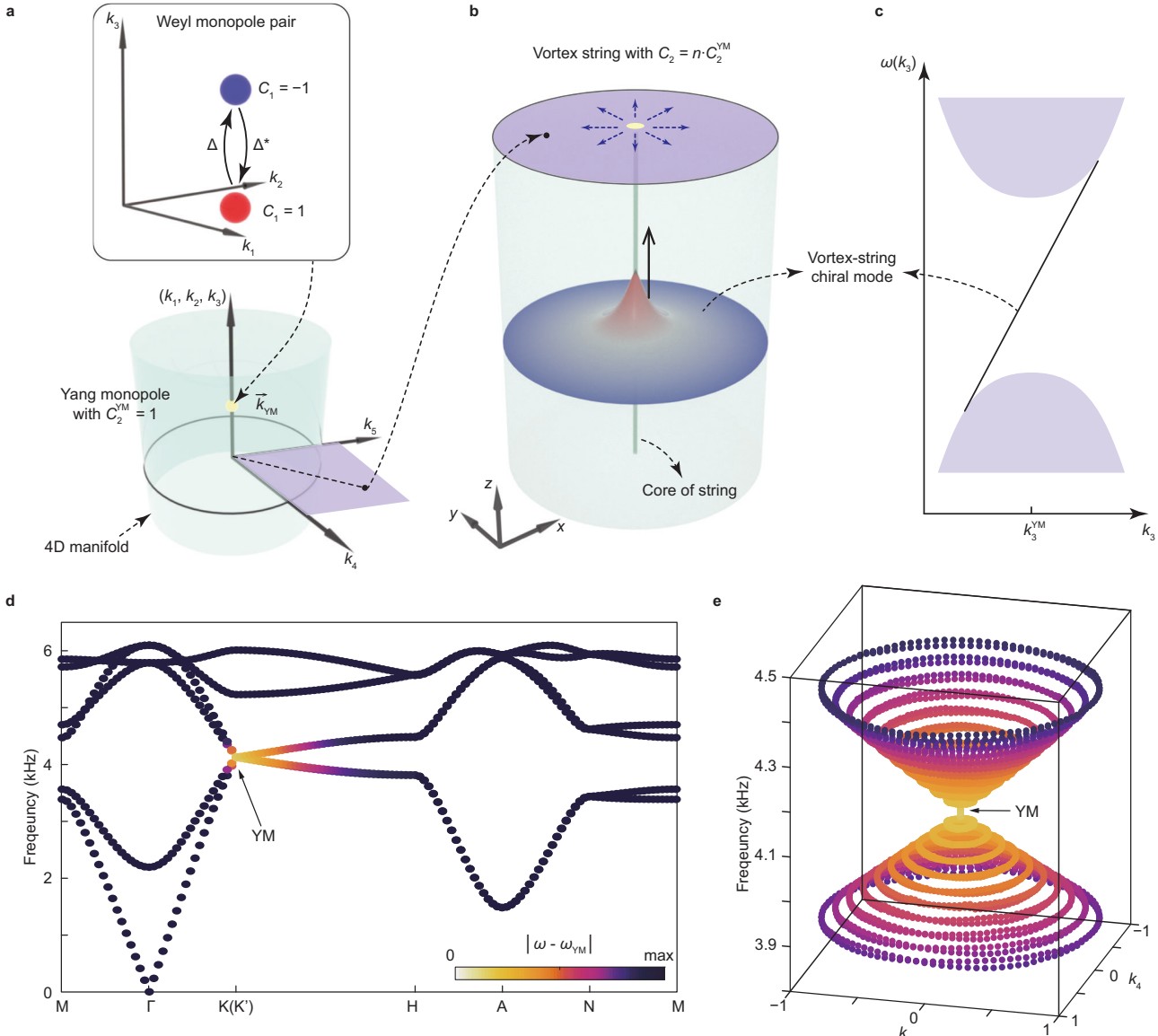

**Fig. 1 | Phononic metamaterials for realising VSC modes. a** Conceptual illustration of a pair of Weyl monopoles with opposite first Chern numbers. If a pair of Weyl monopoles are mixed to each other through a complex scalar field $\Delta = k_4 + ik_5$, they can be considered as a Yang monopole (YM) living in 5D space spanned by three momentum dimensions $(k_1, k_2, k_3)$ and two synthetic dimensions $(k_4, k_5)$. **b** Conceptual illustration of a vortex string. A vortex string is formed when the scalar field $\Delta = k_4 + ik_5$ exhibits nonzero vorticity $n$ around a 1D line in real space. This can be realised by properly defining a mapping from the real space to 2D synthetic space $(k_4, k_5)$. Consequently, VSC modes emerge at the core of the strings. **c** Massless linear dispersion of the VSC modes. This VSC mode is topologically protected by the nonzero second Chern number $C_2$, which originates from the non-Abelian topological charge of YM surrounded by the 4D manifold shown in (**a**). **d** Numerically calculated band diagrams in $(k_1, k_2, k_3)$ momentum space with $(k_4, k_5) = (0, 0)$. **e** Numerically calculated band diagrams in $(k_4, k_5)$ synthetic dimensions with $(k_1, k_2, k_3) = (\pm 4\pi/3a_0, 0, 0)$.

$\theta_0 = \pi/2$. In the experiments, the acoustic wave source is threaded into the bottom or top layer of the device through a round hole near the core of vortex strings (marked by the arrow in Fig. 2c). Figure 2e shows the measured spatial distribution of normalised acoustic pressure along the $z$ direction, suggesting that the acoustic wave can indeed propagate along the core of vortex string with frequency-dependent momentums along $z$ direction. The phase term of the measured acoustic pressure suggests that the frequency for $k_3 = 0$ is 4.16 kHz (Supplementary Fig. 5b), which agrees well with the simulated results of 4.13 kHz. In Fig. 2e, the propagation loss along the $z$-direction has been properly subtracted (Supplementary Note Sec. 2). Figure 2f shows the simulated energy band diagram of the device, which agrees well with the experimentally measured dispersions (Fig. 2g, h), suggesting that such a metamaterial can simultaneously support a pair of

massless VSC modes with opposite non-Abelian topological charge $C_2 = \pm 1$. Note that the coupling between these two modes is negligible in the present scale both numerically and experimentally, as shown in Fig. 2f–h.

Chiral anomaly is a key feature of the VSC modes. Figure 3a, b illustrate the simulated modal profiles at $k_3 = 0.1 \times 2\pi/h$, where the simulated eigenfrequencies of the charge $C_2 = \pm 1$ VSC modes are $\omega(k_3)/2\pi = 4.246$ kHz and $\omega(k_3)/2\pi = 4.020$ kHz, respectively. Figure 3c, d (Fig. 3e, f) show the modal profile of $C_2 = -1$ ($C_2 = 1$) VSC state measured on the third period along positive (negative) $z$ direction at a frequency of 4.246 kHz (4.020 kHz). The simulated and measured modal profiles confirm that the topological modes are tightly confined to the core of the vortex strings, which agrees with the theoretically predicted exponential-decay term along the radial direction in Eq. (1).

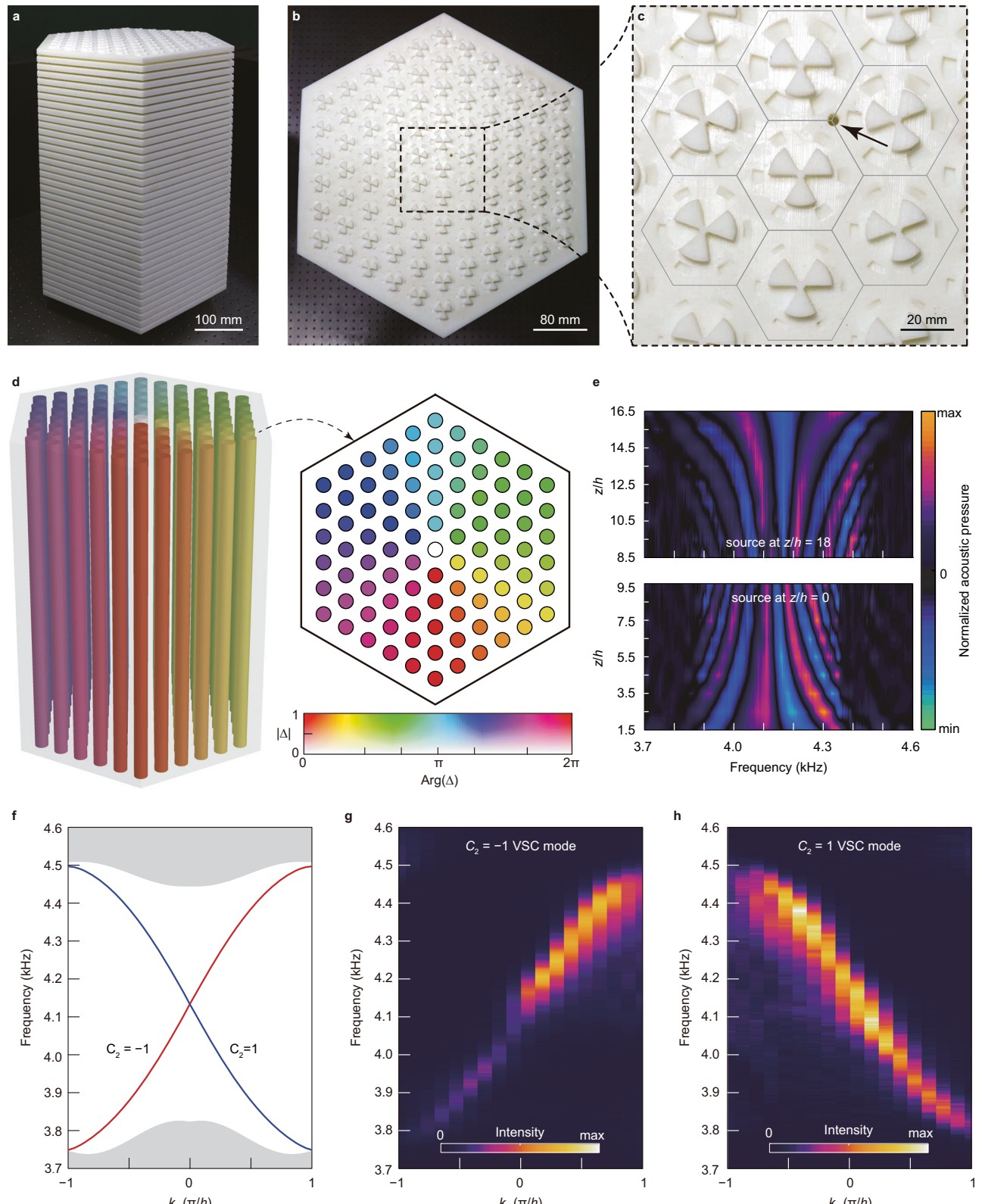

**Fig. 2 | Observation of topological VSC modes in inhomogeneous phononic metamaterials. a** Oblique view of the 3D printed metamaterial device supporting the vortex string. **b** Top view of the experimental device. The entire metamaterial is inhomogeneous in the *x-y* plane. **c** Zoomed-in top-view picture of the structure near the core of the vortex string. A round circle is drilled in the top layer to facilitate efficient excitation of phonons in experiments. **d** Spatial distribution of the complex scalar field $\Delta = k_4 + i k_5$ in the *x-y* plane. The phase of $\Delta$ experience a winding of

$2\pi$ around the centre of the device. **e** Measured spatial distribution of normalized acoustic pressure along *z* direction when the sound source is placed at the top ($z/h = 18$) and bottom ($z/h = 0$) layer respectively. **f** Simulated energy band diagram of the device. There exists a pair of counterpropagating vortex string chiral (VSC) modes with linear dispersions in the bulk bandgap region, which exhibit opposite second Chern numbers $C_2 = \pm 1$. **g, h** Experimentally measured dispersion of VSC with topological charge $C_2 = -1$ (**g**) and $C_2 = 1$ (**h**).

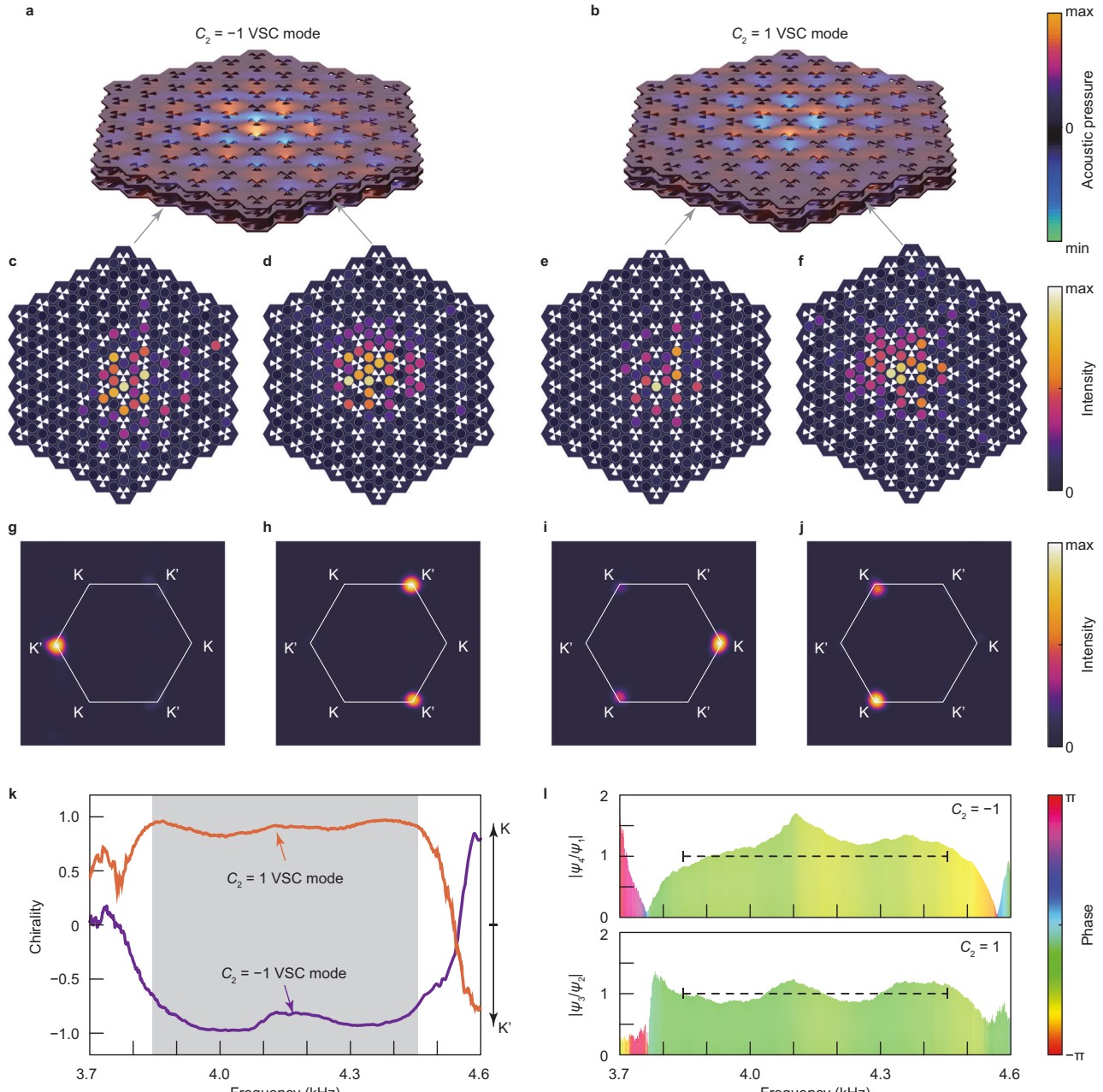

**Fig. 3 | Chiral anomaly behaviours of the topological VSC modes. a, b** Simulated modal profiles at $k_3 = 0.1 \times 2\pi/h$, where the eigenfrequencies of the $C_2 = -1$ (**a**) and $C_2 = 1$ (**b**) vortex string chiral (VSC) modes are $\omega(k_3)/2\pi = 4.246$ kHz and $\omega(k_3)/2\pi = 4.020$ kHz, respectively. **c, d** Modal profiles of $C_2 = -1$ VSC mode measured on the third layer along positive $z$ direction at frequency of 4.246 kHz. **e, f** Modal profile of $C_2 = 1$ VSC mode measured on the sixteenth layer along positive $z$ direction at 4.020 kHz. **g–j** Momentum-space distribution of energy intensity in (**c–f**). The results suggest that the propagating directions of the VSC modes are strictly locked to the K and K' points in the momentum space. **k** Measured chirality

of the forward and backward propagating acoustic wave corresponding to $C_2 = -1$ and $C_2 = 1$ VSC modes. In the bulk bandgap region (grey shaded region), the intensity is mostly near K' (K) point for $C_2 = -1$ ($C_2 = 1$) propagating modes. **l** Measured values of $\psi_4/\psi_1$ and $\psi_3/\psi_2$ suggesting the wavefunction of the VSC modes with $C_2 = -1$ and $C_2 = 1$ respectively. According to Eq. (1), it ideally takes the value of $\exp(-i\pi/2)$. In the bulk bandgap region (marked by dashed line), the amplitude of $\psi_4/\psi_1$ and $\psi_3/\psi_2$ is around unity and the phase is around $-\pi/2$, which agree well with the theoretical prediction.

Most importantly, we project the measured signals into 2D momentum space $(k_1, k_2)$ by conducting spatial Fourier transform and find that the $C_2 = -1$ ($C_2 = 1$) VSC mode locates at the K' (K) point in the 2D momentum space at a frequency of 4.246 kHz (4.020 kHz), as shown in Fig. 3g, h (Fig. 3i, j). Note that the energy distribution in the three equivalent K' (K) points are not equal in each sublayer, which can be explained by expanding the modal profiles of VSC into plane-wave basis (Supplementary Note Section 1.4). Based on the measured data, we analyse the chirality of phonons $(I_K - I_{K'})/(I_K + I_{K'})$ (Fig. 3k),

revealing that the VSC modes exhibit chiral anomaly behaviours, and their propagation directions are determined by the non-Abelian topological charge of YMs at K (K') points.

Based on experimental data, one can also study the four-component wavefunction $(\psi_1, \psi_2, \psi_3, \psi_4)$ of the VSC modes in Fig. 3l. According to Eq. (1), the $C_2 = -1$ VSC mode has four-component wavefunction $(\psi_1, 0, 0, \psi_4)$ where $\psi_4/\psi_1 = \exp(-i\theta_0) = \exp(-i\pi/2)$, and the $C_2 = 1$ VSC mode has wavefunction $(0, \psi_2, \psi_3, 0)$ where $\psi_3/\psi_2 = \exp(-i\theta_0) = \exp(-i\pi/2)$. The measured values of $\psi_4/\psi_1$ and

$\psi_3/\psi_2$ in Fig. 3l slightly deviate in the amplitude, but the phase agrees well with the theoretical prediction. All these results provide conclusive evidence of chiral anomaly propagating behaviours of VSC modes along the vortex strings.

## Discussion

In cosmology, the massless linear dispersion and chiral anomaly of fermionic modes play an important role in making the vortex string "superconducting". These features shown in Figs. 2, 3 are also unique and important in the realm of classical-wave metamaterial systems because they make the vortex string a promising candidate towards practical topological coaxial cables[33] or topological fibres[36]. These features (massless linear dispersion and chiral anomaly) are not available if the 3D metamaterial system is reduced to 2D and the 1D VSC mode is reduced to 0D Dirac-vortex mode[38–41]. These VSC states protected by second Chern numbers are also not available in other 1D line defect studies based on screw dislocations in quasi-3D topological insulators[42,43]. The propagating VSC states are also different from the standing-wave 0D Jackiw-Rebbi defect structure[44]. Besides, our design is also different from the previous theoretical proposals of unidirectional topological fibres in magnetic gyroid photonic metamaterials[36] because our systems can support VSC modes without the necessity of breaking time-reversal symmetry, as far as the two YMs do not mix with each other. It is also important to note that understanding the dynamics of superconducting strings is a crucial aspect of their study in astrophysics. Our current configuration can only simulate static vortex strings. However, given the availability of reconfigurable metamaterials[45,46], we anticipate that it will be possible to investigate dynamics of vortex strings on a time-variant metamaterial platform in future research.

In conclusion, we experimentally construct a vortex string in an inhomogeneous metamaterial and observe the topological VSC modes with massless dispersions and chiral anomaly. Our work not only reveals metamaterials as a useful platform to emulate various cosmic topological phenomena in astrophysics, but also provides a practical strategy for realising topological fibres in the classical-wave systems, finding wide applications in enhancing the robustness of signal processing and communication systems.

## Methods
### Simulations
We used commercial software (COMSOL Multiphysics) to numerically calculate the band structures and Bloch modal profiles, with an air density of $1.18 \, \text{kg/m}^3$ and a sound speed of 343 m/s. In the simulations, air-solid interfaces are assumed to be hard acoustic boundaries. For the calculation of bulk band diagrams in Fig. 1, we used Bloch periodic boundary conditions in all three directions. For the calculation of VSC modes in Figs. 2, 3, we applied Bloch periodic boundary conditions only in the $z$ direction and plane-wave radiation boundary conditions in the $x$ and $y$ directions.

### Experiments
The experimental sample was fabricated with photosensitive resin via stereolithographic 3D printing. For the dispersion measurements in Fig. 2, a broadband sound signal is launched from a balanced armature speaker with a radius of 1 mm, driven by a power amplifier, and located near the core of vortex strings at the top or bottom layer of the sample. Each probe is a microphone (Brüel & Kjær Type 4961) in a sealed sleeve with a tube with a radius of 1 mm and a length of 250 mm. The probes can be threaded into the sample along the horizontal air regions to scan different positions within each layer of the sample. The measured data was processed by a Brüel & Kjær 3160-A-022 module to extract the frequency spectrum with a resolution of 2 Hz. The spatial Fourier transform was applied to complex acoustic pressure signals to obtain the dispersion relation and field distributions.

## Data availability
The data that support the findings of this study are available in the Zenodo repository with the identifier https://doi.org/10.5281/zenodo.10565702.

## Code availability
The codes that support the findings of this study are available in the Zenodo repository with the identifier https://doi.org/10.5281/zenodo.10565702.

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

## Acknowledgements

The work is sponsored by the Key Research and Development Program of the Ministry of Science and Technology under Grants 2022YFA1405200 (YY), No. 2022YFA1404704 (HC), 2022YFA1404902 (HC), and 2022YFA1404900 (YY), the National Natural Science Foundation of China (NNSFC) under Grants No. 62175215 (YY), and No. 61975176 (HC), the Key Research and Development Program of Zhejiang Province under Grant No.2022C01036 (HC), the Fundamental Research Funds for the Central Universities (2021FZZX001-19) (YY), the Excellent Young Scientists Fund Program (Overseas) of China (YY), the National Natural Science Foundation of China under Grant Nos. 12274183 (H-XS) and 12174159 (YG), the National Key Research and Development Program of China under Grant No. 2020YFC1512403 (S-QY), and Hong Kong Research Grants Council under Grant No. N_HKU750/22 (XZ).

## Author contributions

J.M. and Y.Y. conceived the original idea. J.M. consulted Y.Y., H.-X.S., L.Z. and H.C. to design the structures and the experiments. D.J., Y.-J.G., Y.G., H.-X.S. and S.-Q.Y. conducted the experiments. J.M. and Y.Y. wrote the manuscript and interpreted the results with input from all authors. H.-X.S., Y.Y. and X.Z. supervised the project. All authors participated in discussions and reviewed the manuscript.

## Competing interests

The authors declare no competing interests.

## Additional information

**Supplementary information** The online version contains Supplementary Material available at https://doi.org/10.1038/s41467-024-46641-w.

