## [Peer Review File · Nature Communications]

REVIEWER COMMENTS

Reviewer #1 (Remarks to the Author):

A very interesting work, yet not entirely timely since 3D topological defects are at the forefront as for example discussed here:

<https://arxiv.org/abs/2210.09056>

Or in quasi-3D structures comprising screw dislocations discussed in acoustics and photonics settings: Baile Zhang, Zhengyou Liu and Alex Zsameit. The present paper is highly reminiscent of a topological line defect as discussed here

<https://www.nature.com/articles/s41467-018-07817-3>

Now there might be marginal differences, but overall, I see this paper only being incrementally novel and should be submitted to a more specialized journal.

Reviewer #2 (Remarks to the Author):

There paper presents experimental results on the creation of superconducting string analogues in a metamaterial. I have no comments to make on the experimental veracity since this is not my field. I find the idea of interest to those who study superconducting strings in the context of cosmology, but very difficult to understand for a variety of reasons.

After reading the paper and some of the quoted literature the first question I am unclear about is why the material is believed to be described by the Hamiltonian stated in the paper. In addition to that I am unclear about the notation. The identity element - which should be defined in the text - is in bold yet it appears that the bold face is for a 5D vector and the dot product is over the 1-5 indices. My strong recommendation would be to write the expression in index notation so that everything is clear. In the supplementary material they have an expression in which written as $v_{D.(k-k_{YM}).\Gamma}$ - what does this mean. Through the paper the authors have used "." to mean multiplication which is unnecessary and confuses things.

As for what the metamaterial can be described in this way there is a statement in the supplementary material section 2 that "In the tight binding model, ". I presume that this is the explanation which I am looking for. But since I didn't know what this was I felt there should be a reference to a substantive text/review where someone interested in understanding background. Ultimately I am presuming that the behaviour is some kind of mesoscopic description of the material, but under what conditions does it become valid and why are these fulfilled in the material under consideration. Just a cartoon like figure 1 is not really good enough for me.

The key thing that one needs to understand is what is the origin of the scalar field (p_1, p_2). What aspect of the material does it represent and why can it be thought of as some kind of 5D theory in conjunction with the wave number? Moreover is the solution (1) in the main paper a new result from the paper? From the supplementary material section 3, I think it maybe otherwise it needs to be referenced. But in the end what does it represent in terms of the atoms in the material.

The Yang monopole is for a 5D gauge theory. Does (k, p) behave in that way? Why - I think you need to explain rather than just assert.

I think that some of the discussion about topological defects in cosmology is a little simplistic. For example, cosmic strings have been suggested as a source for Large-Scale Structure formation but this is constrained to be at least a subdominant component eg. *Astron. Astrophys.* 571 (2014) A25 • e-Print: 1303.5085 [astro-ph.CO]. I think that the referencing and claims in that respect need to be thought about.

To summarise, I have no way of checking the experimental results, but I think the reason for submission to *Nature Comms* is the connection with areas outside condensed matter/solid state physics. As a potential reader I found the paper difficult to follow and possibly in places assuming too much knowledge of external sources which were often not referenced. I suggest at the very least, substantial rethink.

Reviewer #3 (Remarks to the Author):

The authors report an observation of topological chiral-anomaly states on a vortex string, for the first time, in a carefully constructed experimental set-up. As well as being interesting in its own right, this is also significant as an avenue for experimentally testing an analogue of superconducting cosmic strings - objects that may exist in an astrophysical/cosmological context but have yet to be observed. Having said

that, one of the key aspects of superconducting strings to understand is their dynamics and I believe the construction here prevents the motion of the vortex itself which somewhat limits this approach compared to analogues in a condensed matter context. Nevertheless, the dynamics of currents on the string are still interesting and it is a step forward which, to my knowledge, has yet to be achieved in other contexts.

Comments for the authors:

1). The statement in the abstract about the role of vortex strings in galaxy formation should be softened - the extent of their role is still a matter of debate and, indeed, they may not exist at all.

2). More clarification about how the fields relate to the physical set-up would be useful at an earlier stage. In particular, although it is described in detail later on, a brief description of how the complex scalar field is realised in the experiment would be very helpful shortly after the 5D k vector is introduced (where the 3 momentum vector is discussed but not the two additional synthetic parameters).

3). In the same paragraph, a non-Abelian gauge field is mentioned and it's not clear to me what this refers to. Is this just describing the $SU(2)$ gauge fields in the original context of the YM or is there something in the experimental set-up that can be described as a non-Abelian gauge field, e.g k_{YM} ? If so, the authors should explain what it is explicitly and why it can be described as a non-Abelian gauge field.

4). What are the errors on the measured frequency of 4.13kHz? These should be included in the paper along with any other frequency measurements for comparison to the theory.

5). In figures 3g-j, the VSC mode is clearly located at the K' or K points but sometimes only once and sometimes multiple times. For example, figure 3g has only one point but in 3h the VSC mode appears at both the K' points on the right. Similarly in 3i and 3j, the VSC mode appears twice but with one of the signals at a lesser intensity. It's not clear to me why it appears at some points and not others (sometimes with different intensities) given that there are multiple copies of the K and K' points. Perhaps the authors can expand upon this point.

Apart from my comments above, the figures appear to provide convincing evidence of the authors' claims when compared to the theoretical expectations described in the paper. That said, I am not an expert in acoustic metamaterials so I would recommend seeking the advice of someone more qualified in this regard (if you have not already done so) before publication, given that the quality of the paper crucially depends upon the robustness of the experimental methods.

Response to Reviewer 1

Comment:

A very interesting work, yet not entirely timely since 3D topological defects are at the forefront as for example discussed here: <https://arxiv.org/abs/2210.09056>.

Or in quasi-3D structures comprising screw dislocations discussed in acoustics and photonics settings: Baile Zhang, Zhengyou Liu and Alex Zsameit. The present paper is highly reminiscent of a topological line defect as discussed here: <https://www.nature.com/articles/s41467-018-07817-3>.

Now there might be marginal differences, but overall, I see this paper only being incrementally novel and should be submitted to a more specialized journal.

Response:

We thank the reviewer for bringing these papers to our attention. However, we respectfully disagree with the reviewer's comment regarding the timeliness and novelty of our work. The references provided by the reviewer do not undermine the significance of our work; instead, they exactly highlight its timeliness and novelty.

Our key finding, the vortex-string chiral mode, has been extensively discussed **theoretically** across various physics domains, including astrophysics [Ref. 25-26], condensed matters [Ref. 27–30], and optics [Ref. 31, 33]. The Nature Communication paper mentioned by the reviewer [Ref. 33] is one of them. Despite these theoretical papers, vortex-string chiral mode has not been experimentally observed for decades. Our work, as the first experimental observation of vortex-string chiral modes in metamaterials, is important, timely and novel.

Furthermore, we would like to emphasize that our paper and the references provided by the reviewer have little overlap:

- For example, Prof. Ling Lu's arXiv paper [<https://arxiv.org/abs/2210.09056>] studies the bound state on a 0D Jackiw-Rebbi defect structure, while our focus is the propagating chiral-anomaly state on a 1D vortex-string defect structure, the physics are very distinct. Besides, the topological state in the arXiv paper cannot propagate in any directions and do not exhibit any chiral-anomaly behaviours, while the VSC state in our work is a propagating mode with a well-defined momentum k_3 and exhibits chiral-anomaly behaviours, the observed phenomena are also very different.
- Likewise, the mentioned papers of Prof. Baile Zhang, Zhengyou Liu, and Alex Zsameit [*Nature* 609, 931–935 (2022); *Nat. Commun.* 13, 508 (2022); *Phys. Rev. Lett.* 127, 214301 (2021)] use a screw dislocation to extend the 2D quantum spin Hall systems to quasi-3D systems, while our work is based on 5D Yang monopoles making from 3D Weyl-monopole systems, the physics are fundamentally different. These papers observed screw-defect modes with nonzero first Chern number and zero second Chern number, while our VSC mode exhibit zero first Chern number and nonzero second Chern number, the observed phenomena are irrelevant.

While we disagree with the reviewer's comment, we have added the mentioned references to the manuscript and included discussions clarifying the differences between our work and these papers.

Revision details:

- Main manuscript, the 1st paragraph in 8th page

These VSC states protected by second Chern numbers are also not available in other 1D line defect studies based on screw dislocations in quasi-3D topological insulators^{40,41}. The propagating VSC states are also different from the standing-wave 0D Jackiw-Rebbi defect structure⁴².

- Main manuscript, Refs. 40–42 are added.

Response to Reviewer 2

General Comment:

There paper presents experimental results on the creation of superconducting string analogues in a metamaterial. I have no comments to make on the experimental veracity since this is not my field. I find the idea of interest to those who study superconducting strings in the context of cosmology, but very difficult to understand for a variety of reasons.

General Response:

Thanks for the reviewer's valuable comment. We are delighted to see that the reviewer acknowledges our experimental observation of vortex-string chiral modes in metamaterials, as can be seen from the comment "*the idea of interest to those who study superconducting strings in the context of cosmology*". We also appreciate the reviewer's constructive comment that our previous submission is difficult to understand for researchers outside the metamaterial field. We have added detailed discussions in the revised manuscript, making it accessible for researchers unfamiliar with phononic metamaterials.

In what follows, we will provide detailed point-by-point responses to the comments.

Comment 1.1:

After reading the paper and some of the quoted literature the first question I am unclear about is why the material is believed to be described by the Hamiltonian stated in the paper.

Response:

We agree with the reviewer that it is essential to explain the fundamental principles of metamaterials more effectively to ensure better understanding among the audience.

Let us briefly introduce the concept of "metamaterial". Metamaterials are rationally designed macroscopic structures engineered to respond to external excitations in a tailorable and unconventional fashion. This allows them to achieve properties that are not realizable in conventional materials formed by atomic-scale unit cells. Metamaterials are typically based on periodic arrays of mesoscale artificial unit cells, whose specific shape, size, and geometric orientation give rise to the desired macroscopic properties. Metamaterials make it possible to investigate astronomical phenomena in a controlled laboratory environment. For instance, it has previously been applied to mimic celestial mechanics [Ref. S1]. During the last decades, various topological charges predicted in high-energy physics have also been realized in metamaterials, for instance Weyl [Ref. S2], Dirac [Ref. S3], and Yang monopoles [Ref. S4]. Our work is based on a 5D Yang-monopole Hamiltonian. Similar Hamiltonian has also been realized in an electromagnetic metamaterial very recently [Ref. S4].

The underlying principles of metamaterials exhibiting the proposed Hamiltonian stem from the spatial crystalline symmetry in carefully designed lattice structures. Take graphene as a prime example. Its honeycomb lattice maintains C_6 rotational symmetry, enabling it to support Dirac dispersions near its 2D first Brillouin zone corners [Ref. 35]. This same concept can be applied to phononic metamaterials, with the distinction that acoustic wave equations are solved rather than Schrödinger's equations.

Three-dimensional phononic metamaterials are more complex than the 2D graphene model; however, spatial crystalline symmetry remains crucial for facilitating Weyl/Dirac physics in 3D momentum space. In fact, 3D Weyl-monopole phononic metamaterials have already been reported [Ref. 13]. Compared to Ref. 13, our

contribution is that we used a Brillouin-zone-folding method to construct Yang monopoles from Weyl monopoles. Specifically, we selected a larger unit-cell to make the first Brillouin zone smaller, causing the Weyl monopoles at K (K') and H₀ (H₀') to merge into Dirac monopoles at K (K') point. More interestingly, we discovered that two synthetic parameters k_4 and k_5 , which slightly modify the metamaterials' geometry, can break specific spatial crystalline symmetry and result in a complex mass term $\Delta = k_4 + jk_5$ for the Dirac monopoles. In the 4×4 Dirac equation, k_4 and k_5 play a role like conventional momentum components k_i ($i = 1-3$), allowing us to consider k_4 and k_5 as two additional momentum components and promote the 3D Dirac monopoles to 5D Yang monopoles.

In the revised Supplementary Note, we have provided a systematic and detailed theoretical analysis of phononic metamaterials based on tight-binding approximations. Details of tight-binding approximations will be explained in our response to Comment 2. For better understanding of audiences, we start from the most well-known and simplest 2D graphene-like metamaterial (Supplementary Notes Section 1.1). We compared the theoretical analysis with numerical results simulated from commercial software to confirm that the tight-binding model is an accurate approximation for describing the physics near the Dirac monopoles. Based on this, we moved to 3D phononic metamaterial supporting Weyl monopoles (Supplementary Notes Section 1.2). Then we discussed how the Weyl-monopole metamaterials are promoted to Yang-monopole metamaterials (Supplementary Notes Section 1.3) to support the Hamiltonian proposed in main manuscript.

To facilitate better understanding of audiences, we have also added explanation of the role of k_4 and k_5 at the very early stage of describing YM and vortex strings in the main manuscript. We hope the revised manuscript can satisfactorily address the reviewer's concerns.

Revision details:

- Main manuscript, the 3rd paragraph in 4th page
 “As shown in Fig. 1a, we begin with the discussion of a single YM³. It contains a pair of Weyl monopoles defined in three-dimensional (3D) momentum space (k_1, k_2, k_3) with opposite first Chern numbers $C_1 = \pm 1$...In our experiment, this complex scalar field Δ is introduced by carefully varying the geometry of an ideal Weyl-monopole metamaterials with two independent synthetic parameters k_4 and k_5 (see Supplementary Note Sec. 1). By considering (k_4, k_5) as two synthetic momentum components, the Weyl monopole pair can be upgraded to a YM living in a five-dimensional (5D) space.”
- Supplementary Notes, Section 1, the 1st paragraph
 “Metamaterials, in general, are systematically designed macroscopic structures engineered to have specific responses to external excitations. These responses can be tailored and unconventional, enabling metamaterials to possess properties unattainable in conventional materials made up of atomic-scale unit cells. Metamaterials are typically based on periodic arrays of mesoscale artificial unit cells. These unique properties are typically derived from periodic arrays of mesoscale artificial unit cells, with their specific shape, size, and geometric orientation determining the desired macroscopic characteristics. Metamaterials provide a means to explore astronomical phenomena in controlled laboratory settings. For example, they have been used to simulate celestial mechanics^{S1}. Over recent decades, metamaterials have also been employed to realize various topological charges predicted in high-energy physics, including Weyl^{S2}, Dirac^{S3}, and Yang monopoles^{S4}. This section will focus on the rational design of a phononic metamaterial that exhibits a 5D Yang-monopole Hamiltonian.”
- Supplementary Notes, Sections 1.1–1.3
 Section 1.1 and Fig. S1 are added to analyse the simplest 2D graphene-like metamaterial using tight binding models; Section 1.2 and Fig. S2 are added to investigate a 3D metamaterial supporting Weyl monopoles; Section 1.3 and Fig. S3 are revised to describe the geometry of Yang-monopole

metamaterial and theoretically confirm that such structure can indeed support a Yang-monopole Hamiltonian.

Comment 1.2:

In addition to that I am unclear about the notation. The identity element - which should be defined in the text - is in bold yet it appears that the bold face is for a 5D vector and the dot product is over the 1-5 indices. My strong recommendation would be to write the expression in index notation so that everything is clear. In the supplementary material they have an expression in which written as $v_{D.(k-k_{YM}).Gamma}$ - what does this mean. Through the paper the authors have used "." to mean multiplication which is unnecessary and confuses things.

Response:

We appreciate the reviewer's comment and accept the suggestions to prevent confusion in notation. In the revised manuscript, we have used boldface exclusively for matrices, while vectors are now denoted by italic characters with an arrow above. Additionally, we have presented the expression in index notation to eliminate any ambiguity. The expression " $v_{D.(k-k_{YM}).Gamma}$ " is now written as $\sum_{n=1}^5 v_D(k_n - k_n^{YM})\Gamma_n$, it is a generalized 4-by-4 Dirac Hamiltonian.

Revision details:

- Throughout main manuscript and Supplementary Notes, all vectors are now in italic font with an arrow above, for instance " $\vec{k} = (k_1, k_2, k_3, k_4, k_5) \dots$ " and $\vec{k}_{YM} = (k_1^{YM}, k_2^{YM}, k_3^{YM}, k_4^{YM}, k_5^{YM})$. Matrices are in boldface font, for instance \mathbf{H} and Γ_n ($n = 1-5$). We have use index notation and avoid using dot product in the expression of Dirac Hamiltonian $\mathbf{H} = \omega_{YM}\mathbf{I} + \sum_{n=1}^5 v_n(k_n - k_n^{YM})\Gamma_n$. Note that we have used k_1, k_2, k_3 to represent conventional 3D momentum and k_4, k_5 to represent the two synthetic parameters.

Comment 2:

As for what the metamaterial can be described in this way there is a statement in the supplementary material section 2 that "In the tight binding model, ". I presume that this is the explanation which I am looking for. But since I didn't know what this was I felt there should be a reference to a substantive text/review where someone interested in understanding background.

Ultimately I am presuming that the behaviour is some kind of mesoscopic description of the material, but under what conditions does is this valid and why are these fulfilled in the material under consideration. Just a cartoon like figure 1 is not really good enough for me.

Response:

We appreciate the reviewer's valuable comment, which relates to Comment 1.1.

The reviewer is correct in noting that we theoretically investigate the behaviour of acoustic waves in periodic lattices using the tight-binding approximation, which offers a mesoscopic description of the system. Originally employed in solid-state physics to analyze electronic band structures, the tight-binding approximation assumes that electrons are tightly confined to individual lattice sites and slightly hop to adjacent sites, forming band structures. In our phononic metamaterial, the tight-binding approximation can also be applied to study acoustic

wave band diagrams, provided the free-space acoustic wavelength (~ 85 mm for a Dirac frequency of ~ 4 kHz) is not significantly larger than the lattice constant ($a_0 = 41$ mm). For lower acoustic frequencies (e.g., 1 kHz), the free-space acoustic wavelength (~ 343 mm) becomes too large compared to the lattice constant, rendering the tight-binding approximation invalid, as acoustic waves cannot be tightly bound to a single lattice site. This observation is corroborated by the comparison between tight-binding models and numerical results from commercial software COMSOL in Figs. S1–3.

Our revised Supplementary Note Sec. 1 confirms that the tight-binding approximation accurately describes the underlying physics of our phononic metamaterials near the Weyl/Dirac/Yang monopole frequencies. Additionally, we have included a brief discussion of tight-binding models and relevant references [Ref. 34] in the main manuscript to provide appropriate context for the audience.

Revision details:

- Main manuscript, the 1st paragraph in 6th page
“Based on tight-binding approximation³⁵, we theoretically confirm that such phononic metamaterials can indeed be described by the generalised 4×4 Dirac Hamiltonian...”
- Main manuscript, Ref. 35 is added to explain the tight-binding models.
- Supplementary Notes, Sections 1.1–1.3 and Figs. S1–S3
Section 1.1 and Fig. S1 are added to analyse the simplest 2D graphene-like metamaterial using tight binding models; Section 1.2 and Fig. S2 are added to investigate a 3D metamaterial supporting Weyl monopoles; Section 1.3 and Fig. S3 are revised to describe the geometry of Yang-monopole metamaterial and theoretically confirm that such structure can indeed support a Yang-monopole Hamiltonian.

Comment 3.1:

The key thing that one needs to understand is what is the origin the scalar field (p_1, p_2). What aspect of the material does it represent and why can it be thought of some kind of 5D theory in conjunction with the wave number?

Response:

We appreciate the reviewer’s valuable comment, which relates to Comment 1.1.

In the revised manuscript, we replaced the notation of (p_1, p_2) with (k_4, k_5) . The parameters (k_4, k_5) control the unit-cell geometry of the metamaterial, as illustrated in Fig. S3e. Consequently, different values of (k_4, k_5) correspond to different geometric structures of the unit cells.

Our design is based on Weyl-monopole metamaterials (see Supplementary Note Sec. 1.2 and Fig. S2), which support Weyl monopoles at K (K’) and H_0 (H'_0). In the structure of Fig. S3a, we use a Brillouin-zone-folding method to make the Weyl points at H_0 (H'_0) fold to K (K’), allowing K (K’) to support a pair of Weyl monopoles. Since this structure preserves half-lattice translation symmetry along z direction, the two Weyl points do not mix with each other.

To mix the Weyl monopoles with a complex scalar field $\Delta = k_4 + ik_5$, we need to modify the geometry of the structure from Fig. S3a to Fig. S3e. This modification breaks the half-lattice translation symmetry, enabling the two Weyl monopoles at K (K’) to interact with each other through a nonzero scalar field Δ . We figure out two independent ways to modify the geometry:

- The parameter k_4 controls modification of the spiral air channels so that the dimensions of the spiral channels in A and B substructures shown in Fig. S3e change in opposite directions. As shown in Fig. S3h, the tight-binding model considers this modification as altering the direct and chiral interlayer coupling rate to $3(t_c \pm c_1 k_4)/2$ and $t_c \pm c_1 k_4$. A nonzero value of k_4 breaks the half-lattice translation symmetry along z direction.
- The parameter k_5 controls rotation angle of the fan-shaped scatters in substructure A and B in Fig. S3e. As shown in Fig. S3h, the tight-binding model considers this modification as changing the on-site frequency to $\omega_{\text{YM}} \pm c_2 k_5$. A nonzero value of k_5 simultaneously breaks the half-lattice translation symmetry along z direction and C_2 rotational symmetry in the x - y plane.

The role of (k_4, k_5) has been incorporated into the Hamiltonian Eq. S3. By conducting a Taylor expansion near the $\vec{k}_{\text{YM}} = (k_1^{\text{YM}}, k_2^{\text{YM}}, k_3^{\text{YM}}, k_4^{\text{YM}}, k_5^{\text{YM}}) = (\pm 4\pi/3a_0, 0, 0, 0, 0)$, one can obtain a generalized Dirac Hamiltonian with five Gamma matrices in Eq. S4. We believe that the revised manuscript addresses the reviewer's concerns.

Revision details:

- Main manuscript, the 3rd paragraph in 4th page
“In our experiment, this complex scalar field Δ is introduced by carefully varying the geometry of an ideal Weyl-monopole metamaterials with two independent synthetic parameters k_4 and k_5 (see Supplementary Note Sec. 1). By considering (k_4, k_5) as two synthetic momentum components, the Weyl monopole pair can be upgraded to a YM living in a five-dimensional (5D) space...”
- Main manuscript, the 2nd paragraph in 5th page
“Such YM metamaterials are realized by modifying a previously studied Weyl-monopole metamaterial^{13,14} (see Fig. S2) using a Brillouin-zone-folding method (see Supplementary Notes Sec. 1).”
- Supplementary Notes, Section 1.3, and Fig. S3
The role of (k_4, k_5) has been discussed in detail in this section.

Comment 3.2:

Moreover is the solution (1) in the main paper a new result from the paper? From the supplementary material section 3, I think it maybe otherwise it needs to be referenced. But in the end what does it represent in terms of the atoms in the material.

Response:

Thanks for the reviewer's valuable comment. This equation describes the spatial distribution of the complex scalar field $\Delta = k_4 + ik_5$. We have rewritten the expression in a much simpler form $\Delta(\vec{r}) = k_4 + ik_5 = \Delta_0 e^{in\phi + i\theta_0}$. The phase term $n\phi + \theta_0$ suggests that it has vorticity of n . Similar expression has been studied in a very early paper [Ref. 26], which assumes that the phase of the Higgs field changes by $2n\pi$ in cycling the strings. Note that $\Delta(\vec{r})$ vanishes when $R = 0$. This equation suggests that the parameters (k_4, k_5) are spatially varying. Given that these parameters control the unit-cell geometry of the metamaterial as demonstrated in Fig. S3e, this equation suggests that each atom (or unit cell) at different spatial coordinates $\vec{r} = (R, \phi, z)$ has different geometries. In other words, the entire metamaterial is inhomogeneous, as illustrated in Figs. 2b and d.

Revision details:

- Main manuscript, the 1st paragraph in 5th page

“As shown in Fig. 1, this can be implemented by mapping the spatial-domain cylinder coordinates $\vec{r}=(R, \phi, z)$ to the synthetic subspace (k_4, k_5) so that the scalar field $\Delta(\vec{r}) = k_4 + ik_5 = \Delta_0 e^{in\phi + i\theta_0}$ changes the phase by $2n\pi$ in cycling the strings²⁶. Note that $\Delta(\vec{r})$ vanishes at the origin point $R = 0$. Here, n represents the vorticity of the vortex string, and θ_0 is the phase of scalar field Δ at $\phi = 0$.”

- Main manuscript, Equation 1 is revised.

$$\begin{cases} \psi_- = e^{ik_3 z - \Delta_0 R} \begin{pmatrix} e^{i\theta_0/2} & 0 & 0 & e^{-i\theta_0/2} \end{pmatrix}^T, & \omega(k_3) = \omega_{\text{YM}} + v_{\text{D}} k_3, \\ \psi_+ = e^{-ik_3 z - \Delta_0 R} \begin{pmatrix} 0 & e^{i\theta_0/2} & e^{-i\theta_0/2} & 0 \end{pmatrix}^T, & \omega(k_3) = \omega_{\text{YM}} - v_{\text{D}} k_3. \end{cases}$$

- Supplementary Notes, Section 1.4
Equations S9, S11, and S12 are revised.

Comment 4:

The Yang monopole is for a 5D gauge theory. Does (k, p) behave in that way? Why - I think you need to explain rather than just assert.

Response:

This comment is indeed related to previous comments. We agree with the reviewer’s comment that we need to explain why our system behaves like a Yang monopole in 5D gauge theory. In the revised manuscript, we have included detailed mathematical discussions to confirm that our system can be described by a 5D Yang-monopole Hamiltonian (see Supplementary Notes Sec. 1.3 and Eq. S5). With the same Hamiltonian, we believe that our system exhibits such behaviour.

Revision details:

- Supplementary Notes, Section 1.3
This section is added mainly to explain how to promote 3D Weyl-monopole Hamiltonian to 5D Yang-monopole Hamiltonian using two synthetic parameters (k_4, k_5) . A comprehensive theoretical analysis, grounded in the tight-binding approximation, has been thoroughly examined in this section. Additionally, the updated Figs. S3d and S3h have been provided to enhance the understanding of the discussed concepts.

Comment 5:

*I think that some of the discussion about topological defects in cosmology is a little simplistic. For example, cosmic strings have been suggested as a source for Large-Scale Structure formation but this is constrained to be at least a subdominant component eg. *Astron.Astrophys.* 571 (2014) A25 • e-Print: 1303.5085 [*astro-ph.CO*]. I think that the referencing and claims in that respect need to be thought about.*

Response:

We appreciate the reviewer’s valuable comment. We wholeheartedly agree that our previous discussion about topological defects in cosmology needs revision. In the revised manuscript, we have cited the suggested references [Ref. 21 and 22] and modified our prior discussions on vortex strings in cosmology.

Revision details:

- Main manuscript, 1st sentence of abstract.
“...vortex strings are a potential candidate to influence the formation of the large-scale galaxy clusters that exist today...”
- Main manuscript, 2nd paragraph.
“Vortex strings are postulated to arise from the spontaneous symmetry breaking of the Higgs field during the early stages of the universe. They have been initially proposed as a potential candidate that could play a role in the formation of galaxies in our cosmos²⁰. Continuous efforts have been made to observe vortex strings and set new constraints on their energy scales^{21, 22}, providing valuable insights into the potential existence of these cosmic structures.”
- Main manuscript, Refs. 21 and 22 are added to suggest that recent astronomical observations set certain constraints on cosmic strings.

Comment 6:

To summarise, I have no way of checking the experimental results, but I think the reason for submission to Nature Comms is the connection with areas outside condensed matter/solid state physics. As a potential reader I found the paper difficult to follow and possibly in places assuming too much knowledge of external sources which were often not referenced. I suggest at the very least, substantial rethink.

Response:

We greatly appreciate the reviewer’s constructive comments, which have significantly helped improve our manuscript. In the revised version, we have systematically discussed our design and the underlying theory, progressing from the simplest 2D graphene metamaterial to the 3D Weyl-monopole metamaterial and ultimately our 5D Yang-monopole metamaterial. We have also thoroughly discussed the theoretical analysis, particularly the tight-binding models, in the revised manuscript. We hope this submission addresses the reviewer’s concerns.

Revision details:

- Supplementary Notes, Sections 1.1–1.3 and Figs. S1–S3
Section 1.1 and Fig. S1 are added to analyse the simplest 2D graphene-like metamaterial using tight binding models; Section 1.2 and Fig. S2 are added to investigate a 3D metamaterial supporting Weyl monopoles; Section 1.3 and Fig. S3 are revised to describe the geometry of Yang-monopole metamaterial and theoretically confirm that such structure can indeed support a Yang-monopole Hamiltonian.

Response to Reviewer 3

General Comment:

The authors report an observation of topological chiral-anomaly states on a vortex string, for the first time, in a carefully constructed experimental set-up. As well as being interesting in its own right, this is also significant as an avenue for experimentally testing an analogue of superconducting cosmic strings - objects that may exist in an astrophysical/cosmological context but have yet to be observed. Having said that, one of the key aspects of superconducting strings to understand is their dynamics and I believe the construction here prevents the motion of the vortex itself which somewhat limits this approach compared to analogues in a condensed matter context. Nevertheless, the dynamics of currents on the string are still interesting and it is a step forward which, to my knowledge, has yet to be achieved in other contexts.

General Response:

Thanks for the reviewer's valuable comment. We are delighted to see that the reviewer acknowledges our experimental observation of vortex-string chiral modes in metamaterials, as can be seen from the comment "*being interesting*," "*significant as an avenue for experimentally testing an analogue of superconducting cosmic strings*", "*it is a step forward*".

Our current configuration can only study the dynamics of currents on a static vortex string. The reviewer thinks it is interesting, as can be seen from the comment "*the dynamics of currents on the string are still interesting*". Furthermore, the reviewer raises a fascinating point regarding the potential investigation of the vortex string's dynamics. While this intriguing question goes beyond the present scope of our paper, it certainly merits further exploration in future studies. We appreciate the reviewer's insightful comment.

In what follows, we will provide detailed point-by-point responses to the comments.

Comment 1:

The statement in the abstract about the role of vortex strings in galaxy formation should be softened - the extent of their role is still a matter of debate and, indeed, they may not exist at all.

Response:

Thanks for the reviewer's valuable comment. In the revised abstract, we have softened the statement about the role of vortex strings. Additionally, in the main manuscript, we have also included discussions about the vortex strings in cosmology.

Revision details:

- Main manuscript, 1st sentence of abstract.
"...vortex strings are a potential candidate to influence the formation of the large-scale galaxy clusters that exist today..."
- Main manuscript, 2nd paragraph.
"Vortex strings are postulated to arise from the spontaneous symmetry breaking of the Higgs field during the early stages of the universe. They have been initially proposed as a potential candidate that could play a role in the formation of galaxies in our cosmos²⁰. Continuous efforts have been made to

observe vortex strings and set new constraints on their energy scales^{21, 22}, providing valuable insights into the potential existence of these cosmic structures.”

- Main manuscript, Refs. 21 and 22 are added to suggest that recent astronomical observations set certain constraints on cosmic strings.

Comment 2:

More clarification about how the fields relate to the physical set-up would be useful at an earlier stage. In particular, although it is described in detail later on, a brief description of how the complex scalar field is realised in the experiment would be very helpful shortly after the 5D k vector is introduced (where the 3 momentum vector is discussed but not the two additional synthetic parameters).

Response:

Thanks for the reviewer’s valuable comment. We have added a brief description of how the complex scalar field is experimentally realized before introducing 5D k vectors. Moreover, we have also made significant revisions to the Supplementary Notes to discuss how our metamaterial can exhibit the proposed Hamiltonians. Based on tight-binding approximations, we have systematically discussed our design and the underlying theory, progressing from the simplest 2D graphene metamaterial to the 3D Weyl-monopole metamaterial and ultimately our 5D Yang-monopole metamaterial. We hope this submission addresses the reviewer’s concerns.

Revision details:

- Main manuscript, the 3rd paragraph in 4th page
“In our experiment, this complex scalar field Δ is introduced by carefully varying the geometry of an ideal Weyl-monopole metamaterials with two independent synthetic parameters k_4 and k_5 (see Supplementary Note Sec. 1). By considering (k_4, k_5) as two synthetic momentum components, the Weyl monopole pair can be upgraded to a YM living in a five-dimensional (5D) space...”
- Main manuscript, the 2nd paragraph in 5th page
“Such YM metamaterials are realized by modifying a previously studied Weyl-monopole metamaterial^{13,14} (see Fig. S2) using a Brillouin-zone-folding method (see Supplementary Notes Sec. 1).”
- Supplementary Notes, Section 1.3, and Fig. S3
The role of (k_4, k_5) has been discussed in detail in this section.

Comment 3:

In the same paragraph, a non-Abelian gauge field is mentioned and it's not clear to me what this refers to. Is this just describing the $SU(2)$ gauge fields in the original context of the YM or is there something in the experimental set-up that can be described as a non-Abelian gauge field, e.g k_{YM} ? If so, the authors should explain what it is explicitly and why it can be described as a non-Abelian gauge field.

Response:

Thanks for the reviewer’s valuable comment. The vector \vec{k}_{YM} is the position of YM in 5D space. The non-Abelian gauge field mentioned in the main manuscript is exactly the $SU(2)$ gauge fields in the original context of the YM in Ref. 3. We have revised the relevant sentence to avoid potential confusion among audiences.

Revision details:

- Main manuscript, the 3rd paragraph in 4th page
“Such YM features a SU(2) gauge field in 5D space³. Consequently, it exhibits a non-Abelian second Chern number C_2^{YM} calculated by integrating the curvature of this SU(2) gauge field throughout the four-dimensional (4D) manifold enclosing the YM (see Fig. 1a and Supplementary Note Sec. 1.3).”

Comment 4:

What are the errors on the measured frequency of 4.13kHz? These should be included in the paper along with any other frequency measurements for comparison to the theory.

General Response:

Thanks for the reviewer’s valuable comment. In our previous submission, we used Fig. 2e to estimate the frequency of acoustic waves propagating at $k_3 = 0$. Since Fig. 2e only plots the real part of the complex signals, our previous estimation of frequencies with $k_3 = 0$ is not accurate enough.

In our revised manuscript, we have added the phase term of the complex signals in Fig. S5b to accurately identify the frequency with $k_3 = 0$. The frequencies at which the phase term remains constant along the z direction are 4.153 kHz and 4.166 kHz, obtained from forward and backward propagation measurements respectively. The averaged frequency at $k_3 = 0$ is thus 4.16 kHz, displaying a minor deviation from the simulated results of 4.13 kHz in Fig. 1. Possible reasons for this discrepancy are also discussed.

Revision details:

- Supplementary Notes, Section 2
“Figure 5b shows the phase term of acoustic pressure as a function of frequency and spatial position along z direction. The frequencies at which the phase term remains constant along the z direction are 4.153 kHz and 4.166 kHz, obtained from forward and backward propagation measurements respectively. The averaged frequency at $k_3 = 0$ is thus 4.16 kHz, displaying a minor deviation from the simulated results of 4.13 kHz in Fig. 1. This discrepancy may be attributed to imperfect sample fabrication or slight deviation in temperatures during the measurements. Note that the relative deviation between the measured and simulated frequency is only 0.72%, indicating a strong agreement between experimental and simulation results.”

Comment 5:

In figures 3g-j, the VSC mode is clearly located at the K' or K points but sometimes only once and sometimes multiple times. For example, figure 3g has only one point but in 3h the VSC mode appears at both the K' points on the right. Similarly in 3i and 3j, the VSC mode appears twice but with one of the signals at a lesser intensity. It's not clear to me why it appears at some points and not others (sometimes with different intensities) given that there a multiple copies of the K and K' points. Perhaps the authors can expand upon this point.

Response:

The reviewer raised a very interesting question. To address this question, we need to look at the modal profile of the VSC. Without losing any generality, we focus on VSC mode with $C_2 = 1$. According to Eq. (2), the four-component wavefunction of the VSC state is $e^{ik_3z - \Delta_0 R}(0, e^{i\pi/4}, e^{-i\pi/4}, 0)^T$. The full wave function of the VSC mode is thus $|\text{VSC}\rangle = e^{ik_3z - \Delta_0 R}(e^{i\pi/4}|m_{+, \downarrow}\rangle + e^{-i\pi/4}|m_{-, \uparrow}\rangle)$, where $|m_{+, \downarrow}\rangle$ and $|m_{-, \uparrow}\rangle$ are shown in Fig. S4. Both Bloch modes $|m_{+, \downarrow}\rangle$ and $|m_{-, \uparrow}\rangle$ contain equal component of plane waves at the three equivalent K points (noted by K_1, K_2, K_3). The VSC mode should also contains equal energy component at K_1, K_2, K_3 points. However, since $|m_{+, \downarrow}\rangle$ and $|m_{-, \uparrow}\rangle$ have different distributions in the two sublayers, the components interference with each other so that the energy component at K_1, K_2, K_3 points redistribute in the two sublayers, leading to different patterns shown in Figs. 3g-j.

We expand the VSC mode into a plane-wave basis to mathematically study its components at three equivalent K point. Since $|m_{+, \downarrow}\rangle$ and $|m_{-, \uparrow}\rangle$ are Bloch modes located at K points with vortex-like spatial phase distribution, they can be expanded in a plane-wave basis $|K_n\rangle$ ($n = 1-3$) with three equivalent K points: $|m_{+, \downarrow}\rangle = \frac{e^{i\varphi_{+, \downarrow}}}{\sqrt{3}} \sum_{n=1}^3 e^{i2(n-1)\pi/3} |K_n\rangle$ and $|m_{-, \uparrow}\rangle = \frac{e^{i\varphi_{-, \uparrow}}}{\sqrt{3}} \sum_{n=1}^3 (-1)^l e^{-i2(n-1)\pi/3} |K_n\rangle$. Here l represents the sublayer index along z direction. Note that the additional phase terms $i\varphi_{+, \downarrow}$ and $i\varphi_{-, \uparrow}$ are determined by the specific Gamma matrix of the Dirac equation, and they do not influence the mode profiles of $|m_{+, \downarrow}\rangle$ and $|m_{-, \uparrow}\rangle$. One can thus expand the VSC mode in a plane-wave basis:

$$|\text{VSC}\rangle = \frac{e^{ik_3z - \Delta_0 R}}{\sqrt{3}} \sum_{n=1}^3 \left(e^{i\varphi_{+, \uparrow} + i\theta_0/2 + i2(n-1)\pi/3} + (-1)^l e^{i\varphi_{-, \downarrow} - i\theta_0/2 - i2(n-1)\pi/3} \right) |K_n\rangle.$$

One can clearly find that VSC mode can have different components at the three K points. This explains the inequivalent energy distribution in three K points in Fig. 3g-j. Very interestingly, if we sum up the total energy distribution of the two sublayers, we can find that the VSC mode still exhibit equal energy distribution in the three equivalent K points with a constant value of $\sum_{l=\pm 1} \left| e^{i\varphi_{+, \uparrow} + i\theta_0/2 + i2(n-1)\pi/3} + (-1)^l e^{i\varphi_{-, \downarrow} - i\theta_0/2 - i2(n-1)\pi/3} \right|^2 = 4$. We have added relevant discussions into the revised manuscript.

Revision details:

- Main manuscript, 8th page
“Note that the energy distribution in the three equivalent K’ (K) points are not equal in each sublayer, which can be explained by expanding the modal profiles of VSC into plane-wave basis (Supplementary Note Sec. 1.4).”
- Supplementary Notes, Section 1.3, 8th page
“These Bloch wavefunctions $|m_{\pm, \uparrow\downarrow}\rangle$ can also be expanded plane-wave basis $|K_n\rangle$ ($n = 1-3$) with three equivalent K points wavevectors

$$\begin{aligned} |m_{\pm, \downarrow}\rangle &= \frac{e^{i\varphi_{\pm, \downarrow}}}{\sqrt{3}} \sum_{n=1}^3 e^{i2(n-1)\pi/3} (\pm 1)^l |K_n\rangle \\ |m_{\pm, \uparrow}\rangle &= \frac{e^{i\varphi_{\pm, \uparrow}}}{\sqrt{3}} \sum_{n=1}^3 e^{-i2(n-1)\pi/3} (\pm 1)^l |K_n\rangle \end{aligned}$$

Here $l = \pm 1$ denotes different substructures A and B in Fig. S3e. One notes that the additional phase term $\varphi_{\pm, \uparrow\downarrow}$ in these expressions is determined by the specific choice of the five Gamma matrices.”

- Supplementary Notes, Section 1.4, 11th page

“Note that Eq. (S11) operates on Bloch modes $|m_{\pm, \uparrow\downarrow}\rangle$ shown in Supplementary Fig. 4, which can be further expanded into plane wave basis using Eq. (S6), one can obtain the mode profile of the VSC mode:

$$|\text{VSC}\rangle = \frac{e^{ik_3 z - \Delta_0 R}}{\sqrt{3}} \sum_{n=1}^3 \left(e^{i\varphi_{+, \uparrow} + i\theta_0/2 + i2(n-1)\pi/3} + (-1)^l e^{i\varphi_{-, \downarrow} - i\theta_0/2 - i2(n-1)\pi/3} \right) |K_n\rangle,$$

which suggests that the component of three K point is no longer equal in the VSC mode if we only look at plane-wave expansions in different sublayers with $l = \pm 1$. If we sum up the total energy distribution of the two sublayers, we can find that the VSC mode has equal energy distribution in the three K points with $\sum_{l=\pm 1} \left| e^{i\varphi_{+, \uparrow} + i\theta_0/2 + i2(n-1)\pi/3} + (-1)^l e^{i\varphi_{-, \downarrow} - i\theta_0/2 - i2(n-1)\pi/3} \right|^2 = 4$ independent from for the values of $n = 1 - 3$.”

Comment 6:

Apart from my comments above, the figures appear to provide convincing evidence of the authors' claims when compared to the theoretical expectations described in the paper. That said, I am not an expert in acoustic metamaterials so I would recommend seeking the advice of someone more qualified in this regard (if you have not already done so) before publication, given that the quality of the paper crucially depends upon the robustness of the experimental methods.

Response:

We greatly appreciate the reviewer’s constructive comments, which have significantly helped improve our manuscript. We have revised the manuscript according to the reviewer’s suggestions, and we believe all reviewer’s concerns have been addressed.

REVIEWERS' COMMENTS

Reviewer #2 (Remarks to the Author):

I think that paper is now much more easy for me to understand and connect with things which are more standard the topological defect community. With my previous caveat that I am unable to comment much on the experimental aspects of the paper - which are the key results - I am happy to recommend its publication as something which I think is of significant general interest and could lead to interest within the cosmological community working on topological defects.

I have one further issue with the text. I am not happy with the very first statement. As I explained in my previous report current observations mean that topological defects are unlikely to be the most dominant mechanism for galaxy formation. The paper currently makes this statement without reference and indeed I think it is misleading. My suggestion is that they say something along the lines of "topological defects could have played many roles in cosmology" and reference the book by Vilenkin and Shellard which is the standard text in this context. When they talk about specific constraints around refs 21,22 I think that they should also mention the paper by the Planck collaboration *Astron.Astrophys.* 571 (2014) A25 • e-Print: 1303.5085 [astro-ph.CO] since it is the most relevant.

Reviewer #3 (Remarks to the Author):

The updated version has been improved in line with my suggestions which I believe has made the paper more easy to understand, particularly the connection between the theory and the experimental set-up.

I still have some difficulty understanding the details as to why the system can be understood as containing a YM monopole in 5D space. However, I'm willing to believe that this may be due to my lack of understanding of metamaterials given that the authors have made a reasonable attempt to explain it in more detail in the supplementary notes.

Since the paper has the potential to be of interest to other fields, I would be willing to overlook these concerns and recommend it for publication, if it is also approved by a referee with specialisation in this area.

Response to Reviewer 2

Comment:

I think that paper is now much easier for me to understand and connect with things which are more standard the topological defect community. With my previous caveat that I am unable to comment much on the experimental aspects of the paper - which are the key results - I am happy to recommend its publication as something which I think is of significant general interest and could lead to interest within the cosmological community working on topological defects.

*I have one further issue with the text. I am not happy with the very first statement. As I explained in my previous report current observations mean that topological defects are unlikely to be the most dominant mechanism for galaxy formation. The paper currently makes this statement without reference and indeed I think it is misleading. My suggestion is that they say something along the lines of "topological defects could have played many roles in cosmology" and reference the book by Vilenkin and Shellard which is the standard text in this context. When they talk about specific constraints around refs 21,22 I think that they should also mention the paper by the Planck collaboration *Astron.Astrophys.* 571 (2014) A25 • e-Print: 1303.5085 [astro-ph.CO] since it is the most relevant.*

Response:

Thanks for the reviewer's valuable comment. We are delighted to see that the reviewer acknowledges the significance of our work, as can be seen from the comment "*recommend its publication*", "*of significant general interest and could lead to interest within the cosmological community working on topological defects*".

We also agree with the reviewer's constructive comment that we should be careful about the statement of cosmic strings and its references. We have revised the first statement of cosmic strings and added the suggested references into the revised manuscript.

Revision details:

- Main manuscript, abstract
"As hypothetical topological defects in the geometry of spacetime, vortex strings could have played many roles in cosmology, and their distinct features can provide observable clues about the early universe's evolution."
- Main manuscript, Refs. 20 and 24 are added.

Response to Reviewer 3 ----- NCOMMS-23-29996A

Comment:

The updated version has been improved in line with my suggestions which I believe has made the paper more easy to understand, particularly the connection between the theory and the experimental set-up.

I still have some difficulty understanding the details as to why the system can be understood as containing a YM monopole in 5D space. However, I'm willing to believe that this may be due to my lack of understanding of metamaterials given that the authors have made a reasonable attempt to explain it in more detail in the supplementary notes.

Since the paper has the potential to be of interest to other fields, I would be willing to overlook these concerns and recommend it for publication, if it is also approved by a referee with specialisation in this area.

Response:

Thanks for the reviewer's valuable comments to make our manuscript better. We are delighted to see that the reviewer acknowledges the significance of our work, as can be seen from the comment "*has the potential to be of interest to other fields*", "*recommend it for publication*".